# Further Evidence of the Melatonin Calmodulin Interaction: Effect on CaMKII Activity

**DOI:** 10.3390/ijms23052479

**Published:** 2022-02-24

**Authors:** Jesús Argueta, Héctor Solís-Chagoyán, Rosa Estrada-Reyes, Luis A. Constantino-Jonapa, Julián Oikawa-Sala, Javier Velázquez-Moctezuma, Gloria Benítez-King

**Affiliations:** 1Laboratorio de Neurofarmacología, Subdirección de Investigaciones Clínicas, Instituto Nacional de Psiquiatría Ramón de la Fuente Muñiz, Calzada México-Xochimilco 101, San Lorenzo Huipulco, Tlalpan, Mexico City 14370, Mexico; jadclear@yahoo.com (J.A.); hecsolch@imp.edu.mx (H.S.-C.); biologia0712@gmail.com (L.A.C.-J.); oikawasala@gmail.com (J.O.-S.); 2División de Ciencias Biológicas y de la Salud, Posgrado en Biología Experimental, Universidad Autónoma Metropolitana, Iztapalapa, Mexico City 09340, Mexico; 3Laboratorio de Fitofarmacología, Dirección de Investigaciones en Neurociencias, Instituto Nacional de Psiquiatría Ramón de la Fuente Muñiz, Calzada México-Xochimilco 101, San Lorenzo Huipulco, Tlalpan, Mexico City 14370, Mexico; restrada@imp.edu.mx; 4Clínica de Trastornos del Sueño, Universidad Autónoma Metropolitana, Iztapalapa, Mexico City 09340, Mexico; jvm@xanum.uam.mx

**Keywords:** calmodulin, melatonin, calmodulin kinase II, colocalization, hippocampus

## Abstract

Melatonin (MEL) is a pleiotropic indolamine that reaches multiple intracellular targets. Among these, MEL binds to calmodulin (CaM) with high affinity. In presence of Ca^2+^, CaM binds to CaM-dependent kinase II (CaMKII). The Ca^2+^-CaM/CaMKII pathway regulates a myriad of brain functions in different cellular compartments. Evidence showing the regulation of this cellular pathway by MEL is scarce. Thus, our main objective was to study the interaction of MEL with CaM and its effects on CaMKII activity in two microenvironments (aqueous and lipidic) naturally occurring within the cell. In addition, colocalization of MEL with CaM in vivo was explored in mice brain hippocampus. In vitro CaM-MEL interaction and the structural conformations of CaM in the presence of this indoleamine were assessed through electrophoretic mobility and isoelectric point. The functional consequence of this interaction was evaluated by measuring CaMKII activity. Ca^2+^-CaM-MEL increased the activity of CaMKII in aqueous buffer but reduced the kinase activity in lipid buffer. Importantly, MEL colocalizes in vivo with Ca^2+^-CaM in the hippocampus. Our evidence suggests that MEL regulates the key cellular Ca^2+^-CaM/CaMKII pathway and might explain why physiological MEL concentrations reduce CaMKII activity in some experimental conditions, while in others it drives biological processes through activation of this kinase.

## 1. Introduction

Melatonin (N-acetyl-5-methoxytryptamine, MEL, indoleamine) is mainly synthesized by the pineal gland (at night, dark phase) reaching 10^−7^ M in the cerebrospinal fluid; moreover, neuronal, and glial cells synthesize MEL at specific brain regions reaching a local concentration of 10^−5^ M [1]. In addition, MEL is synthesized in peripheral organs such as the skin, among others [2,3,4]. MEL binds to receptors MT1 and MT2 in the plasma membrane and to the third melatonin receptor MT3 quinone reductase in the cytosol [5]. Due to its amphiphilic nature, MEL can permeate the membrane by passive diffusion [6,7]. Intracellularly, MEL can have a modulatory effect on some organelles, particularly on the mitochondria, which could be related to redox and bioenergetic homeostasis [2]. In addition, MEL acts on targets such as calcium-binding proteins such as calmodulin (CaM) and calretinin [8], distributed in the cytosol and internal membranes (aqueous or lipidic compartments, respectively) [9]. Particularly, MEL can bind with high affinity to purified CaM, in Ca^2+^ dependent fashion, integrated into liposomes [10].

CaM is distributed in cytosolic and membranal subcellular compartments [11,12] where it orchestrates a great variety of key functions. CaM is formed by 148 amino acids (with 35 acidic and 15 basic residues) [13] and binds four Ca^2+^ to be activated [14]. Activated-CaM (Ca^2+^-CaM or holo-CaM) adopts a dumbbell-shaped conformation with two globular EF-hands domains joined by an alpha helix structure. In contrast, in the absence or chelation of Ca^2+^ (EGTA-CaM or apo-CaM), CaM has a random globular conformation [15]. MEL binds to the hydrophobic cleft in both domains of the dumbbell-shaped CaM [16,17]. CaM conformations are related to concentration and time of exposure to Ca^2+^ ions and to its binding proteins. Upon binding to its targets, CaM modifies and settles in certain conformations favoring the affinity to those targets and activating specific pathways to finally achieve important cellular responses [18,19,20]. To drive its cellular functions, Ca^2+^-CaM binds to a broad spectrum of proteins to regulate their structural or enzymatic functioning. An important target of Ca^2+^-CaM is the CaM-dependent kinase II (CaMKII), an enzyme that plays a key role in neurodevelopment, a process that also occurs in the adult brain. CaMKII is broadly distributed but primarily accumulates in the hippocampus, a brain structure where new neurons develop and differentiate to be integrated to preexistent neuronal circuitries. CaMKII and its substrates [21] are found in the cytosol, but also associated with the cytoskeleton in intracellular membranes [22] to drive dendritogenesis and synaptogenesis. Interestingly, evidence suggests that the interaction of MEL with CaM influences the activity of CaMKII, i.e., the in vitro activity of purified CaMKII was diminished by MEL treatment in a concentration-dependent manner [23].

In this paper we assessed the spatial overlap of MEL and CaM labeled with specific antibodies in a brain region which is a target for MEL. In addition, we studied CaM conformations formed in lipidic and aqueous microenvironments in presence of MEL as well as their functional effects on CaMKII activity.

## 2. Results

### 2.1. Colocalization of Melatonin with Calmodulin In Vivo

To study whether MEL presents spatial overlap with CaM in vivo, mice were administered 16 mg/kg MEL at the dark phase, when high levels of MEL are present. Brain hippocampal slices were simultaneously incubated with anti-MEL and anti-CaM antibodies, followed by secondary antibodies coupled to different fluorochromes. As shown in Figure 1, scarce stained spots of MEL (red) were observed in the hippocampal slices of mice treated with the vehicle (VEH) (Figure 1A). In contrast, in the presence of MEL, abundant spots of CaM and MEL were detected (Figure 1B). Some colocalization sites could be observed as yellow spots (white arrows) (Figure 1B). Colocalization assessment showed that mice treated with MEL presented 77% (mean = 10.9, SEM = 1.50) more colocalization sites than those treated with the VEH (mean = 2.5, SEM = 0.98) (Figure 1C). Nested *t* test showed no differences between colocalization sites in samples from mice with the same treatment. Of all the colocalization sites detected, a Pearson’s coefficient of about 0.5 or more was considered a valid colocalization site of CaM and MEL signals. These results suggest that CaM and MEL spatially overlap in mice hippocampus, and thus, interactions between these two molecules might occur in vivo.

### 2.2. Determination of Optimal Electrophoretic Mobility of Calmodulin with and without Calcium

Relative mobility (Rf) of CaM was assessed by electrophoretic separation in denaturing conditions (SDS-PAGE) and in conditions where the native CaM conformation was preserved (Native-PAGE). SDS-PAGE separation of CaM (Figure 2A) showed a minimal non significative difference in the Rf of Ca^2+^-CaM (mean = 0.838, SEM = 0.010) and EGTA-CaM (mean = 0.833, SEM = 0.016). In contrast, native-PAGE separation of CaM (Figure 2B) showed a difference of nearly 16% (* *p* < 0.05) between the Rf of Ca^2+^-CaM (mean = 0.719, SEM = 0.029) and EGTA-CaM (mean = 0.854, SEM = 0.031). Thus, we chose the native-PAGE method to study the interaction between MEL and CaM.

### 2.3. Melatonin Modifies Calmodulin Conformation: Assessment by Two-Dimensional Separation

To further characterize conformational CaM changes induced by MEL in aqueous or lipidic microenvironments, we incubated this protein with Ca^2+^ and either 10^−7^ M MEL or the VEH and then resolved it by native 2D gel electrophoresis.

The master gel included the CaM spots formed in all experimental conditions (aqueous or lipidic microenvironment with or without Ca^2+^). These spots in the master gel had a 95% similarity among the analyzed gels and only those consistently present in all gels (intra and inter-assay) were included. In the master gel 9 spots were found, i.e., spots that were significantly different when compared to the global signal (master gel) (Figure 3A).

The 9 spots with an isoelectric point (IP) range between pH of 3.5 to 4.5 were used to generate a heatmap and a dendrogram shown in Figure 3B. Spot data in the dendrogram were shown as the standard deviation with respect to the global mean of the master gel (Z-score). Spots 1, 2 and 3 of CaM were the most abundant in lipidic microenvironment in presence of Ca^2+^ and MEL, while very low formation of these spots was observed in samples incubated in aqueous microenvironment with EGTA and without Ca^2+^. In contrast, spot 4 was the most abundant in medium without Ca^2+^ (Figure 3B). Moreover, CaM incubated with Ca^2+^ and MEL in aqueous microenvironment showed decreased intensity in spots 1, 2 and 5 to 9 (Figure 3B). In lipidic microenvironment, Ca^2+^-CaM without MEL showed high intensity in spots 1 and 2. In the presence of the indoleamine, the intensity of these spots slightly diminished while it increased in spot 3 (Figure 3B). Interestingly, the conditions (aqueous or lipidic) that included Ca^2+^-CaM-MEL were grouped into opposite different clusters (Figure 3B).

Representative images of CaM incubated in aqueous (Figure 3C) or lipidic buffers (Figure 3D) separated by 2D electrophoresis are shown. Spots were observed between pH 3.7 and 4.7 and with Rf from 0.37 to 0.65. Importantly, the spots had different fluorescence intensity according with the microenvironment and the presence of MEL. Outstandingly, the Ca^2+^-CaM-MEL interaction in aqueous buffer displaced the IP of some spots by 2 pH units (from 4.5 to 6.5) suggesting that Ca^2+^-CaM adopted a different stable conformation when medium was close to the cytosolic pH (Figure 3C). In addition, CaM incubated without Ca^2+^ and with the Ca^2+^ chelating agent EGTA, induced changes in fluorescence intensity of the CaM spots in lipidic environment (Figure 3D).

These data suggest that the incubation of Ca^2+^-CaM with MEL induced the formation of reproducible spots, presumable by stable molecular interactions which were influenced by the microenvironment.

To support 2D results showing that Ca^2+^-CaM-MEL complexes present different conformations we study the UV-vis absorbance spectrum (See method in supplementary document). Spectra of Ca^2+^-CaM and Ca^2+^-CaM incubated with MEL were measured from 250 nm to 350 nm (Appendix A). The spectra of Ca^2+^-CaM showed a peak from 260 nm to about 280 nm as previously reported [24]. Changes of Ca^2+^-CaM-MEL absorbance occur in comparison to the Ca^2+^-CaM alone. The spectra of Ca^2+^-CaM and Ca^2+^-CaM-MEL were tested at 3.5, 4.5, 6.5 and 7.5 values (Appendix A). At pH 4.5, no changes in the absorbance were observed after addition of MEL. However, at pH 3.5, 6.5 and 7.5 Ca^2+^-CaM-MEL absorbance diminished when compared to Ca^2+^-CaM. Together these results suggest that MEL could influence conformational changes in Ca^2+^-CaM.

### 2.4. The CaM-MEL Interaction up- or Downregulates CaMKII Activity According with the Microenvironment

To test whether the interaction between CaM and MEL modifies the activity of CaMKII, we measured its enzymatic activity in vitro, in both aqueous and lipidic buffers.

The activity of CaMKII activated by Ca^2+^-CaM tested in aqueous buffer showed a 2-fold increase compared to the activity of the enzyme measured in the absence of Ca^2+^ (Figure 4A). EGTA-CaM had a minimum activity (mean = 46,519, SEM = 12,704) compared with the Ca^2+^-CaM activity in aqueous (mean = 443,671, SEM = 12,704) and lipidic (mean = 205,251, SEM = 37,757) buffers. CaMKII activity determined in the aqueous media in presence of 10^−7^ M MEL (Figure 4B) was increased by 31.6% in comparison to the activity of the enzyme in the presence of VEH (Figure 4A). While in the lipidic microenvironment, the activity of CaMKII diminished by 70% in the presence of 10^−7^ M MEL (Figure 4B). These results suggest that CaM conformations formed in the presence of MEL in aqueous microenvironment upregulate CaMKII activity, while those formed in the lipidic microenvironment downregulate CaMKII activity.

## 3. Discussion

Evidence from nuclear magnetic resonance-based studies and ligand-binding assays have shown that MEL can bind to Ca^2+^-CaM [10,16,17]. Exogenous MEL antidepressant-like effects in mice have been recently described [25,26], and modulation of neuroplasticity may underlie this effect. CaMKII plays a key role in neurogenesis and dendritogenesis which are processes stimulated by antidepressants and MEL [27]. In this study, we showed for the first time that MEL colocalizes with CaM in hippocampal slices of MEL-treated mice. The relevance of this finding is that MEL could become close enough to CaM and this spatial overlap raises the possibility of a molecular interaction between them. In addition, our evidence supports that CaM adopted different spatial conformations in presence of MEL in aqueous and lipidic microenvironments, differentially modulating the activity of CaMKII. By modulating CaMKII activity, the CaM-MEL interaction might influence cellular processes such as the ones involved in neuroplasticity.

Regarding the assessment of CaM-MEL colocalization, the immunofluorescent detection methodology does not allow us to discern whether the individual immunoreactive signals correspond to either free MEL or CaM, or to MEL bound to CaM. Considering that MEL is a small molecule, it is possible that MEL attached to CaM cannot be recognized by its antibody due to steric hindrance. Moreover, the microscope resolution is not enough to detect a simple CaM-MEL interaction. However, the data obtained from this methodology help us to identify regions where CaM-MEL spatial closeness occur with high possibility. Then, all the colocalization sites suggest regions where CaM-MEL could be interacting. It was demonstrated previously that dendrite formation is induced in rodent hippocampal slices incubated with MEL. This process involves the participation of CaMKII [28] by an independent MT receptor mechanism. Herein we showed that mice injected with a dose of MEL that produces an antidepressant-like behavior and dendrite formation [25] had abundant colocalization sites of CaM and MEL in the hippocampus, suggesting that a molecular interaction might take place in vivo.

Tridimensional structure of CaM has been extensively studied by several methods [18,19,29,30]. These studies point out that CaM has a high degree of conformational plasticity, as it adopts different sub-states [31] induced principally by its binding to Ca^2+^ and target proteins [14,32]. CaM conformations have also been studied by electrophoretic separation in native gels [19,29,30] because proteins mobility in an electrical field depends on the electrostatic driving force, the hydrodynamic drag, amino acid composition, charge, shape, mass as well as features of the microenvironment such as pH and viscosity. Importantly, previous evidence indicates that pH gradient can induce changes in CaM [33] and that the electrophoretic mobility of Ca^2+^-CaM and EGTA-CaM is different in aqueous buffers as it occurred in this work and depends on changes in the molecular conformation [34,35,36]. Thus, based in these facts we characterized the conformational changes of Ca^2+^-CaM and EGTA-CaM by 2D electrophoresis in native gels and we found that MEL induced stable modifications on the conformation of Ca^2+^-CaM both in aqueous and lipidic microenvironments. Lipidic buffer was formed by phosphatidylcholine which constitutes liposomes, a membrane-like experimental model used to study lipidic membrane-dependent protein activity [37].

As a control, we separated CaM purified from bovine brain by SDS-PAGE in denaturing conditions. A unique band with the expected molecular weight of 16 kDa was observed (Figure 2), indicating that electrophoretic mobility shift was not due to protein degradation. Thus, the heterogeneous electrophoretic mobility of CaM observed in our diverse experimental conditions can be explained by the interaction of CaM with Ca^2+^ and/or MEL and by the influence of the microenvironment.

Moreover, shifts in IP and MW were observed after 2D-electrophoresis corroborating that MEL induces different proportions of CaM conformations; one conformation being more abundant (IP = 3.5) in lipidic microenvironment than in aqueous media. In the same sense, in the presence of MEL Ca^2+^-CaM adopted 5 conformations of similar IP (4.7) and different molecular weights. Differences in Rf on native 1D-electrophoresis has been related principally to the protein size (given by conformation or molecular weight) and net charge. On the other hand, protein mobility in isoelectric focusing is related to conformational changes that expose charged amino acids along a pH gradient as well as to net charge or size. Therefore, in both 1D- and 2D-electrophoresis, the relative mobility of the purified native CaM observed in this study can be associated with stable conformational changes adopted by this protein when it interacted with Ca^2+^ and MEL.

Regarding the interaction between CaM and MEL observed in this study, the following findings can be highlighted: (1) the greatest effect on IP of Ca^2+^-CaM interaction with MEL in aqueous buffer; (2) the almost neutral changes in conformations of CaM in absence of Ca^2+^ in aqueous buffer; (3) the accumulation of some CaM conformations in the presence of MEL in lipidic buffer.

Some structural and physicochemical properties of CaM allow us to explain the 9 stable conformations of CaM-MEL found in our study. At least 45 three-dimensional structures of CaM have been documented [38]. Moreover, constructs of CaM with two fluorescent dyes introduced into the opposing EF-hands have demonstrated the simultaneous formation of distinct sub-states of this protein in aqueous buffer [29]. These sub-states are tightly related to variations in size, with changes that go from the closed shape in EGTA-CaM to the open, activated and dumbbell shape adopted by CaM after Ca^2+^ binding [39]. In addition, antagonists of CaM, also produce huge conformational changes of Ca^2+^-CaM upon binding [40]. Therefore, our results about the interaction of Ca^2+^-CaM with MEL in aqueous buffer suggest that this indoleamine can deeply influence (maybe as a stabilizer) the conformation of CaM adopted under this condition. UV-vis spectroscopy was applied to support the 2D-electrophoresis results. Data obtained strongly suggest that pH-dependent CaM conformational changes occur when it is incubated with MEL, supporting that MEL influences the conformational plasticity of CaM.

The biological relevance of the interaction between the most physiologically versatile Ca^2+^ -binding protein and MEL in aqueous or lipidic microenvironments was tested in vitro and in vivo. In vitro, through evaluating the activity of one important CaM target, the enzyme CaMKII. Interestingly, different effects of MEL on the conformation of Ca^2+^-CaM facilitated by the microenvironment concur with the kinase activity. In aqueous buffer we observed an increase in the activity of CaMKII, whereas in a lipidic microenvironment the interaction of Ca^2+^-CaM and MEL induced a decreased activity. These data agree with the previous reported effects in which the in vitro CaMKII activity was inhibited by MEL, whereas in hippocampal brain slices the activity of this enzyme was stimulated. The effect of MEL in aqueous microenvironment might be explained by enhancement of stability in the EF-hands, which are important domains involved in the binding of Ca^2+^-CaM to CaMKII. In addition, enhancement of the kinase activity might be due to an increase in the CaM affinity for Ca^2+^, consequently increasing the stability of the Ca^2+^-CaM complex [41]. On the other hand, the decrease in the CaMKII activity [23] in lipidic buffer might be explained by a possible reduction of CaM affinity for Ca^2+^, and the stability of the Ca^2+^-CaM/CaMKII complex. However, further research is needed to confirm this hypothesis.

## 4. Materials and Methods

### 4.1. Reagents

To perform assays, we used a highly purified calmodulin (CaM) from bovine brain (Calbiochem, 208694, lot. D00161715, San Diego, CA, USA), melatonin (MEL) (Sigma Aldrich, M5250), carbonic anhydrase (Sigma Aldrich, C-2273, St. Louis, MO, USA), EGTA (Sigma Aldrich, E4378), CaCl_2_ (Sigma Aldrich, C1016), phosphatidylcholine (Sigma Aldrich, P2772) and deionized Milli-Q^®^ water (Millipore, Direct-Q3 UV). In addition, SYPRO^TM^ ruby protein blot stain (ThermoFisher Scientific, S11791, Waltham, MA, USA), SYPRO^TM^ ruby protein gel stain (ThermoFisher Scientific, S21900), low fluorescence PVDF membranes (Millipore, IPFL00010, Burlington, MA, USA), anti-CaM antibody (Sigma, C7055) for Western blot, anti-CaM antibody (Abcam, ab2860, Cambridge, UK) for immunofluorescence, anti-MEL antibody (Serotec, 0100-0203, Oxford, UK), Alexa Fluor^®^ 488 anti-mouse IgG (Invitrogen, A32723, Waltham, MA, USA), DyLight^TM^ 488 anti-mouse IgG (SA5-10166) and DyLight^TM^ 680 anti-rabbit IgG (Invitrogen, SA5-10042). For kinase activity assessment, ADP-Glo kinase assay (Promega, V6930, Madison, WI, USA) with CaMKIIγ enzyme system (Promega, V3531) was used.

### 4.2. Colocalization of Calmodulin and Melatonin

In vivo interaction of MEL with CaM was studied by colocalization of CaM and MEL in mice hippocampus. Mice experimental procedures were approved by the Ethics Committee from the Instituto Nacional de Psiquiatría Ramón de la Fuente Muñiz, project number NC19127.0, following the specifications of Mexican Official Norm (NOM-062-ZOO-1999) and laboratory animal care (NIH publication # 85-23, revised in 1985). Male Swiss Webster mice (25–35 g) were housed in groups of 8 per cage (44 cm × 21 cm × 21 cm) in a temperature-controlled room (20–21 °C). Animals had free access to food and water during the whole experiment. Mice were housed in a 12 h inverted light/dark cycle (ZT 0 = lights on; 12 h light/12 h dark) and were gathered into two groups: One was treated with the VEH and the other with MEL. Administration of MEL (16 mg/kg) was assessed by a three times dose injection of the VEH or MEL. First injection was administered 24 h before the sacrifice, and the second and third injections were administered 11 h and 30 min [26] before the anesthesia administration. Mice were perfused with paraformaldehyde 2% with CaCl_2_ 0.02%, and finally embedded in sucrose for cryopreservation [42].

Brains were obtained and coronal brain tissue slices were cut with a cryotome (Microm HM 25). Slices thickness were about 6 μm, mounted in glass slides, incubated with a mouse anti-calmodulin antibody (1:50) followed by a rabbit anti-melatonin antibody (1:50). Tissue was incubated overnight at 4 °C and finally, primary antibodies were detected with DyLight^TM^ 488 anti-mouse IgG (1:250) and DyLight^TM^ 680 anti-rabbit IgG (1:250) for 60 min at room temperature. Preparations were mounted with ProLong Diamond (ThermoFisher Scientific, P36961). Cured for 24 h and then analyzed in a LSM 700 confocal microscope (Carl Zeiss microscopy, Jena, Germany). Two colocalization controls were analyzed, one without anti-calmodulin antibodies, and the second without anti-melatonin antibodies; both controls were incubated with secondary anti-rabbit IgG and anti-mouse IgG antibodies. This was carried out to rule out nonspecific colocalization between the two secondary antibodies and to ensure that the colocalization corresponds to MEL and CaM.

Data were gathered from 2 slices obtained from the hippocampus of 2 mice injected with VEH. Two random images of each slice were acquired. For MEL treatment, data were obtained by a similar procedure from 3 mice. Results were analyzed with a nested *t* test.

Images were acquired with a 63× magnification and a 4× digital zoom at 512 × 512 pixels with 0.05 μm for x and y parameters and for the z interval 1.0 μm [43]. Raw images were modified with the deconvolution non-linear least-square algorithm in the ImageJ plugin DeconvolutionLab2. Deconvolved images were analyzed for colocalization sites with an object-based analysis in the ImageJ plugin JACoP [44].

### 4.3. Separation and Detection of Calmodulin

CaM (0.5 µg) was resolved by SDS polyacrylamide gel electrophoresis (PAGE) in gradient gels (4–20% polyacrylamide) for Ca^2+^ (0.5 mM CaCl_2_) and EGTA (0.5 mM) CaM in running buffer (25 mM Trizma base, 192 mM glycine and 3 mM SDS). Transferred to PVDF membranes and blotted with a primary anti-calmodulin antibody (1:500) at 4 °C overnight. Finally, membranes were incubated with a secondary Alexa Fluor^®^ 488 anti-mouse antibody (1:15,000) for 20 min. Images were acquired with a ChemiDoc MP (BIO-RAD, Hercules, CA, USA) imaging system and analyzed (relative front (Rf) calculation) with the Image Lab software version 5.2.1 (BIO-RAD).

### 4.4. Native 2D Electrophoresis

CaM was incubated for 15 min at room temperature with 10 mM CaCl_2_ (Ca^2+^) or 10 mM EGTA and each condition either with VEH or 10^−7^ M MEL for 15 min at room temperature, in either aqueous (10 mM Trizma pH 7.4) or lipidic (10 mM Trizma pH 7.4 and 200 μg of phosphatidylcholine) buffers. Then CaM was separated by native two-dimensional (2D) gel electrophoresis [45]; the first dimension was resolved in IPG strips (BIO-RAD, 1632000) with ampholytes of pH ranging from 3 to 10. The second dimension was performed in gradient polyacrylamide gels (4–20%) and later stained for CaM with SYPRO^®^ ruby. Images were acquired with the ChemiDoc MP system. 2D protein spots were statistically analyzed (one-way ANOVA and Bonferroni’s test) with the DECODON DELTA2D software ver. 2.8.2, Demo version. The significant spots were analyzed to generate a heatmap and dendrogram with the R software ver. 3.6.1.

### 4.5. CaMKII Enzymatic Activity

CaMKIIγ activity was assessed with the ADP Glo assay kit. The phosphorylation assay was performed following the instructions given by the supplier. CaM (5 ng/μL) was purchased from Calbiochem. Enzymatic reaction was achieved with 0.5 mM CaCl_2_, or 0.5 mM EGTA, Tris-buffer (20 mM Tris, 10 mM MgCl_2_ and BSA 1 mg/mL), Autocamptide-2 (KKALRRQETVDAL-amide, derived from the autophosphorylation site (amino acids 283–290) on CaMKII) 0.2 μg/μL, CaMKII 0.6 ng/μL and 50 μM ATP. To determine the effect of the CaM-MEL interaction, reactions were carried out for 15 min at room temperature adding the VEH or increasing MEL concentrations (10^−11^, 10^−9^, 10^−7^, 10^−5^ M), in aqueous or lipidic buffers (80 ng/μL of phosphatidylcholine). Chemiluminescence was assessed in the ChemiDoc MP imaging system (BIO-RAD).

### 4.6. Statistical Analysis

All experiments were carried out at least by triplicate, except for the 2D native gel electrophoresis experiments, which were performed by duplicate. Comparisons (where they were necessary) were carried out by a two-sample *t* test, a nested *t* test, a one-way or two-way analysis of variance (ANOVA) followed by Tukey’s test. In all analysis, a *p* value < 0.05 was considered significant. Graphs and statistical analysis were performed using GraphPad Prism 8 for Windows, (GraphPad Software, San Diego, CA, USA) Some of the graphic panels were performed with the software Affinity Publisher, ver. 1.10.4.1198.

## 5. Conclusions

In this study we showed several CaM conformations adopted during its interaction with MEL. The proportion of these conformations varied depending on aqueous or lipidic nature of the microenvironment. Interestingly, Ca^2+^-CaM-MEL interaction differentially modulates the activity of CaMKII, increasing the activity of this kinase in aqueous buffer or inhibiting its phosphorylating activity in lipidic environment. Both aqueous and lipidic microenvironments are components of the inner structure of the cells and MEL, the Ca^2+^-binding protein CaM and their targets are broadly distributed in aqueous and lipidic compartments. Therefore, they may interact with their respective targets or substrates under the influence of these environments after a transitory increase in the cytosolic Ca^2+^ concentration. In vivo colocalization of MEL with CaM occurs in the hippocampus of mice injected with MEL with a dose that causes an antidepressant-like effect. This result suggests that interaction between MEL and CaM could happen in vivo. Altogether, evidence contributes to explain why MEL antagonizes CaMKII activity in some experimental conditions, while in others the indoleamine drives biological processes through activation of this enzyme. Thus, the results of our study suggest that various conformations of Ca^2+^ binding proteins such as CaM may participate in eliciting multiple cellular processes in response to MEL signaling.

## Figures and Tables

**Figure 1 ijms-23-02479-f001:**
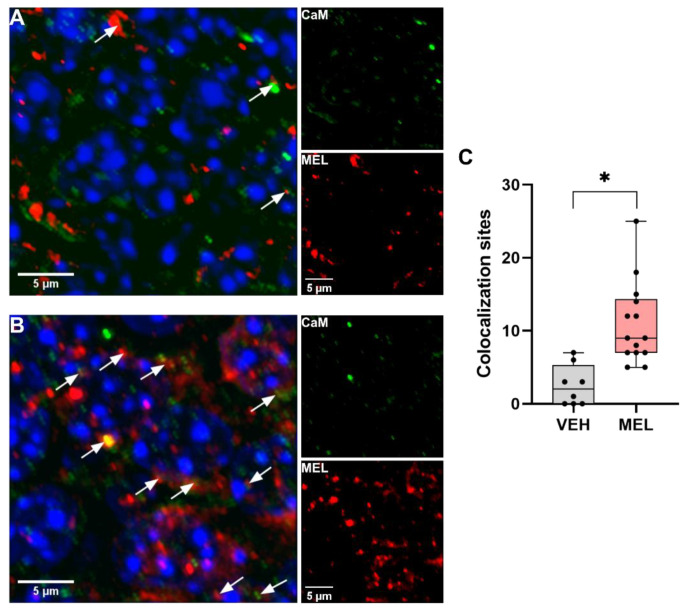
Colocalization of calmodulin and melatonin in the brain hippocampus. Mice were injected with the vehicle or 16 mg/kg melatonin (MEL) as described in Methods. Fixed brain hippocampal coronal slices were stained with specific anti-calmodulin and anti-melatonin antibodies by indirect immunofluorescence. (**A**) Representative image of the calmodulin (CaM) and MEL colocalization sites in mice treated with the vehicle (VEH) and (**B**) with MEL. Colocalization sites are identified with white arrows in the merged image. Single signals images for CaM (green) and MEL (red) are shown. Blue signals correspond to DAPI-stained cellular nuclei. (**C**) Colocalization sites in hippocampal zones were assessed in the slices of mice treated with the VEH or MEL. The graph shows data obtained from 2 mice injected with the VEH or MEL. Data were acquired from 2 near hippocampal zone slices, and per slice 2 random hippocampal images were taken. For MEL treatment, data were obtained from 3 mice from which 2 near hippocampal zone slices were obtained, and 2 random hippocampal images per slice were taken. Results were analyzed with a nested *t* test, * *p* < 0.05 was significant.

**Figure 2 ijms-23-02479-f002:**
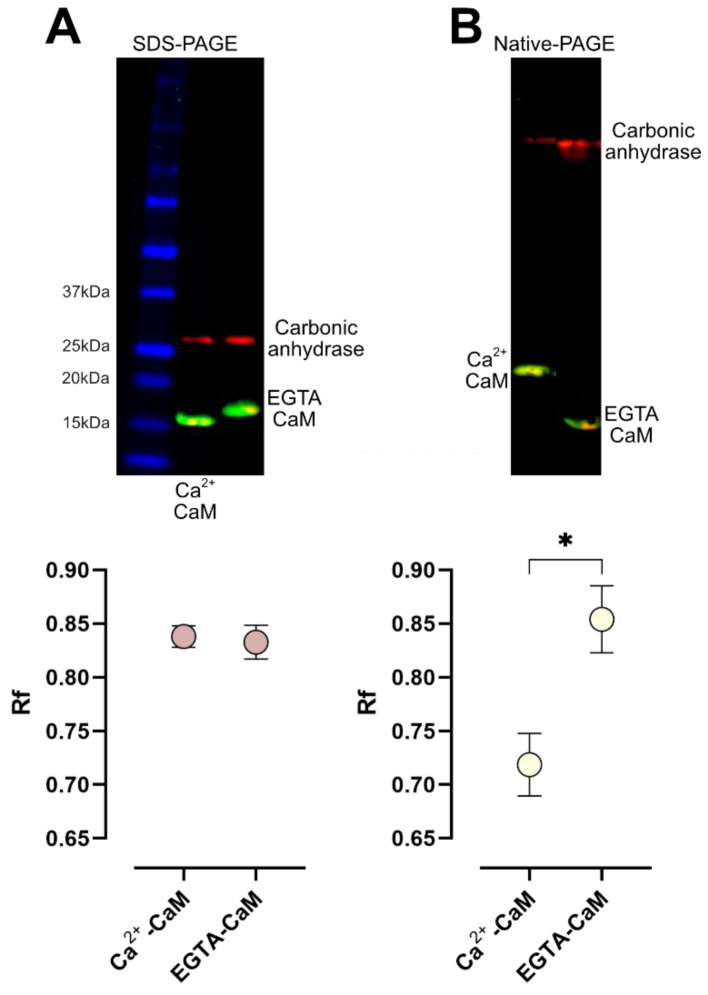
Effect of Ca^2+^ and EGTA in the relative mobility of calmodulin. (**A**) SDS-PAGE of calmodulin (CaM) pre-incubated with either Ca^2+^ or EGTA. (**B**) Native-PAGE of CaM pre-incubated with either Ca^2+^ or EGTA. In both cases, carbonic anhydrase was included as external loading control. Spot graphs show the relative mobility (mean, SEM) of the denatured (SDS-PAGE) or the native (Native-PAGE) CaM. Data of 3 independent experiments were analyzed with a two-sample *t* test, * *p* < 0.05.

**Figure 3 ijms-23-02479-f003:**
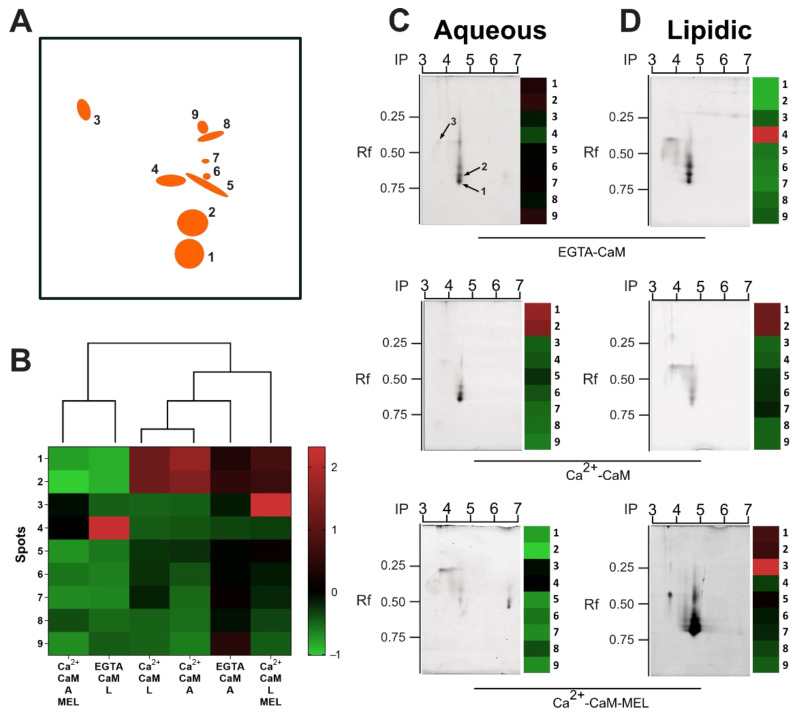
Calmodulin separation by native 2D electrophoresis. Calmodulin (CaM) was incubated with calcium (Ca^2+^-CaM), or with EGTA (EGTA-CaM), in presence of the vehicle (VEH) or with 10^−7^ M melatonin (MEL) in either aqueous or lipidic buffers for 10 min. CaM was detected by SYPRO^®^ Ruby stain and gels were run at least 2 times. (**A**) Significant spot summary of CaM conformations formed in a lipid or aqueous microenvironment are shown. (**B**) Results of the spot differences between the experimental groups are shown in the heatmap as Z-scores (standard deviation of the global mean), zero means no changes with respect to the mean of spots within the master gel. Representative images of CaM separated by 2D gel electrophoresis are shown. (**C**) Spot conformations of CaM in aqueous solution. (**D**) Spot conformations of CaM in lipidic solution. A: aqueous or L: lipidic buffers. Data were analyzed by a one-way ANOVA test and a Bonferroni multiple comparisons test. Significant differences were considered with *p* < 0.05.

**Figure 4 ijms-23-02479-f004:**
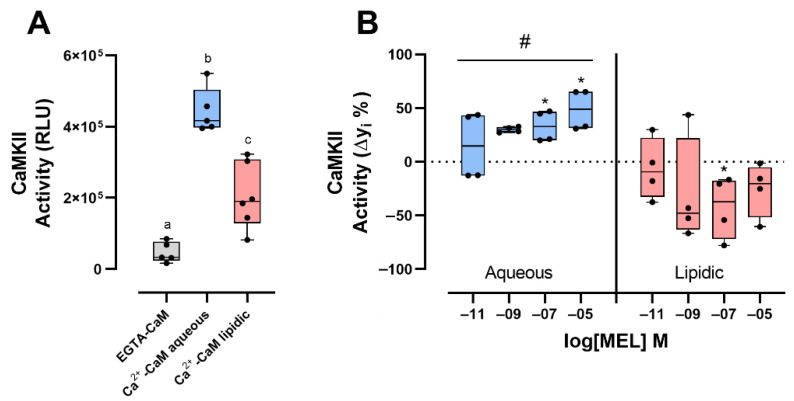
Calmodulin kinase II activity determined in the presence of melatonin. (**A**) Basal calmodulin kinase II (CaMKII) activity in the presence of calmodulin (CaM) with EGTA or Ca^2+^ in aqueous or lipidic microenvironment. (**B**) Percentage of the difference ($\Delta y_{i}\%$) between CaMKII basal activity and the activity of CaMKII assessed in the presence of Ca^2+^-CaM and four melatonin (MEL) concentrations (10^−11^, 10^−9^, 10^−7^, 10^−5^ M) in aqueous or lipidic media. Zero is the activity of CaMKII determined with Ca^2+^-CaM and the vehicle (Illustrated as a punctuated line). A one-way or two-way ANOVA and a Tukey multiple comparisons test were performed. Significant differences (*p* < 0.05) between the vehicle (aqueous or lipidic) and different MEL concentrations are indicated with (*), and (#) indicates global differences between aqueous and lipidic environments. Different literals *p* < 0.05. Results are represented as box and whiskers with the median, first and third quartile with maximum and minimum values of at least 4 experimental repetitions.

## Data Availability

The data presented in this study are available on request from the corresponding author.

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
