# Peer review of "Further Evidence of the Melatonin Calmodulin Interaction: Effect on CaMKII Activity"

_ijms, 2022, doi:10.3390/ijms23052479_

Round 1
Reviewer 1 Report
In the current study, the authors have reported that further evidence of the melatonin calmodulin interaction: effect on CaMKII activity. This is a straightforward study and it provides some new evidence related to melatonin and calmodulin interaction which is a controversial topic for years. Several issues should be addressed to further improve the quality of the report.
- In the current study, the melatonin antibody was used to identify its location. If melatonin was bound with its antibody, it would not bind to calmodulin any more. Thus, the colocalization of melatonin/antibody with calmodulin is non-specific and meaningless. The melatonin injection definitely will increase the non-specific colocalization compared to the control as indicated in the figure 1. This issue should be addressed in the text and a reasonable explanation should be provided.
- It is mentioned in text that “quinone reductase in the plasma membrane”. This is incorrect. Quinone reductase is located in the cytosol.
- The protocol number for animal study should be provided.
- It should specify what is the activity of CaMKII, i.e., what is the substrate of CaMKII used in this study and also the product? This important information is absent in the text.
Reviewer 2 Report
In the manuscript entitled “Further evidence of the melatonin calmodulin interaction: Effect on CaMKII activity” the Authors show the interaction of melatonin (MEL) with calmodulin (CaM) in vivo in mouse brain hippocampal slices. They also show, by 2D electrophoresis, that MEL might have an effect on the conformation of CaM depending on the microenvironment (aqueous and lipidic). Since MEL regulates the Ca2+/CaM/CaMKII pathway the Authors analyzed the activity of CaMKII in the presence CaM or MEL-CaM.
Major points:
- Conformations of CaM in the presence of MEL, in aqueous and lipidic microenvironment, should be analyzed/confirmed by an additional method (for instance biophysical /spectroscopic one).
- Manuscript represents poor English, so its revised version requires careful language correction.
Additional points (only examples):
Since Authors use the term melatonin and indoleamine interchangeably they should mention in Introduction that melatonin is an indoleamine; they should not use term indoleamine in Abstract.
Abbreviations “CaM”, “MEL”, “Ca2+” should be used throughout the entire manuscript; in many places (Figure legends, subtitles) terms “calmodulin”, “melatonin” or “calcium” are used.
Lane 45 – instead of “35 acid” should be “35 acidic”.
Lane 73 – term “VEH” should be explained since it appears here for the first time.
Lane 100 – instead of significative” should be “significant”.
Lane 175 – Instead of “A) Basal CaMKII and calmodulin (CaM) activity in presence of EGTA or Ca2+ in aqueous or lipidic microenvironment” should be “A) Basal CaMKII activity in the presence of CaM with EGTA or Ca2+ in aqueous or lipidic microenvironment”.
Reviewer 3 Report
This is an interesting paper, which requires some revisions.
While I do not have a critique as relates to methodology, my recommendations relates to the background and data interpretation.
Introduction, first paragraph.
Melatonin is produced in many peripheral organs (Cell Mol Life Sci 74(21), 3913-3925, 2017), which should be indicated.
The authors mention nuclear receptors without specifying them. Note, RORA is not a receptor for melatonin as documented recently.
As relates to overview of receptors for melatonin cf. (Mol Cell Endocrinol 351:152-66), 2012).
Indicate that melatonin can affect cellular functions by acting on mitochondria.
Also indicate that phenotypic effects can be indirect through metabolites (Exp Dermatol 26:563–568, 2017), since melatonin metabolism in peripheral tissues can be very rapid (FASEB J 27, 2742–2755, 2013).
Please better emphasize novelty since it is already known the melatonin acts on calmodulin.
Round 2
Reviewer 1 Report
The authors have not properly answered the question that the reviewer raised, i.e., whether after melatonin binds to calmodulin it still can band to melatonin antibody or most importantly, whether after melatonin binds to its antibody, it still can bind to calmodulin. Without clarifying this important issue in the text, the data are not reliable.
Reviewer 2 Report
The revised version of the manuscript entitled “Further evidence of the melatonin calmodulin interaction: Effect on CaMKII activity” has been only partially corrected. The Authors still use the term indoleamine interchangeably with melationin and do not explain why.
Also, the abbreviations “CaM”, “MEL”, “Ca2+” are not used consequently throughout the entire manuscript. Moreover, the Reviewer indicated only some examples and Authors should find by themselves other not proper terms and sentences and correct them. So, the manuscript still represents poor English.
Regarding the major point, the Reviewer is convinced that at least one additional method should be applied (measurement of fluorescence, CD spectra?) to support the final conclusion.
Reviewer 3 Report
The authors adequately revised the manuscript
Author Response
Nothing to change, thank you.
Round 3
Reviewer 2 Report
To improve the English style and clarity of the inserted text (line 162-171) the Reviewer proposes to substitute it for the following one:
”To support 2D results showing that Ca2+-CaM-MEL complexes present different conformations we study the UV-vis absorbance spectrum (See method in supplementary file). Spectra of Ca2+-CaM and Ca2+-CaM incubated with MEL were measured from 250 nm to 350 nm (Figure 1s). The spectra of Ca2+-CaM showed a peak between 260 nm and about 280 nm as previously reported [24]. Changes of Ca2+-CaM-MEL absorbance occur in comparison to the Ca2+-CaM alone. The spectra of Ca2+-CaM and Ca2+-CaM-MEL were tested at 3.5, 4.5, 6.5 and 7.5 pH values (Figure 2s). At pH 4.5, no changes in the absorbance were observed after addition of MEL. However, at pH 3.5, 6.5 and 7.5 Ca2+-CaM-MEL absorbance diminished when compared to Ca2+-CaM. Together, these results suggest that MEL could influence conformational changes in Ca2+-CaM.”
Author Response
Corrections were made as suggested in the results section.